# Toward Semantic History Compression for Reinforcement Learning

**Fabian Paischer** [1], **Thomas Adler** [1], **Andreas Radler** [1], **Markus Hofmarcher** [2], **Sepp Hochreiter** [1] [3]

[1] ELLIS Unit Linz and LIT AI Lab, Institute for Machine Learning,
[2] JKU LIT SAL eSPML Lab, Institute for Machine Learning,
Johannes Kepler University, Linz, Austria
[3] Institute of Advanced Research in Artificial Intelligence (IARAI), Vienna, Austria
`paischer@ml.jku.at`

## Abstract

Agents interacting under partial observability require access to past observations via a memory mechanism in order to approximate the true state of the environment. Recent work suggests that leveraging language as abstraction provides benefits for creating a representation of past events. History Compression via Language Models (HELM) leverages a pretrained Language Model (LM) for representing the past. It relies on a randomized attention mechanism to translate environment observations to token embeddings. In this work, we show that the representations resulting from this attention mechanism can collapse under certain conditions. This causes blindness of the agent to subtle changes in the environment that may be crucial for solving a certain task. We propose a solution to this problem consisting of two parts. First, we improve upon HELM by substituting the attention mechanism with a feature-wise centering-and-scaling operation. Second, we take a step toward semantic history compression by leveraging foundation models, such as CLIP, to encode observations, which further improves performance. By combining foundation models, our agent is able to solve the challenging MiniGrid-Memory environment. Surprisingly, however, our experiments suggest that this is not due to the semantic enrichment of the representation presented to the LM, but rather due to the discriminative power provided by CLIP. We make our code publicly available at `https://github.com/ml-jku/helm`.

## 1  Introduction

In Reinforcement Learning (RL) an agent interacts with an environment and learns from feedback provided in the form of a reward function. RL agents that are deployed in the real world often have to cope with partial observability. Therefore, the capability to approximate the true state of their surrounding environment by virtue of an agent's perception is crucial (Åström, 1964; Kaelbling et al., 1998). To this end, many agents employ a memory mechanism to track events that occurred in the past. In this memory, it is much more efficient to store abstract representations of the past rather than every detail the agent encountered. Thus, memory mechanisms such as LSTM (Hochreiter & Schmidhuber, 1997) or Transformer (Vaswani et al., 2017) compress sequences of high-dimensional observations.

Similarly, also humans memorize abstract concepts rather than every detail of information they encountered in the past (Richards & Frankland, 2017; Bowman & Zeithamova, 2018). In humans, the ability to abstract is heavily influenced by the exposure to language in early childhood (Waxman & Markow, 1995). By design, language is very well suited to form *abstractions* and it is used on a

Language and Reinforcement Learning Workshop at Neural Information Processing Systems, 2022.

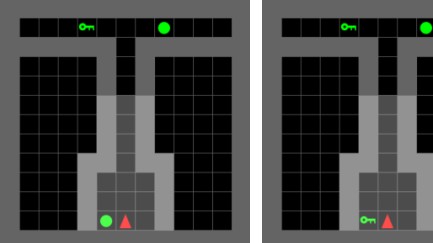

| $\beta$ | Hellinger distance $\delta$ |
|---|---|
| 1 | $0.00036 \pm 1.88\text{e-}9$ |
| 10 | $0.0036 \pm 1.84\text{e-}7$ |
| 100 | $0.035 \pm 2.09\text{e-}5$ |
| 1000 | $0.091 \pm 0.01$ |
| 1e4 | $0.093 \pm 0.06$ |

Figure 1: **Left:** Instances of MiniGrid-Memory environment. The agent only observes the shaded region, and must navigate to the object seen in the starting room. **Right:** Hellinger distance (Hellinger, 1909) of softmax distributions over token embeddings for the two observations of the Memory environment on the left. Mean over 1000 different initializations of $\boldsymbol{P}$ of FH is shown.

daily basis to pass on information between humans. Therefore, it is a natural choice as a medium for compressing compounding information. Prior work has illustrated that pretrained LMs can efficiently compress sequences of observations and facilitate agent learning in partially observable RL environments (HELM, Paischer et al., 2022). The key challenge hereby is to map image-based observations to language representations. HELM tackles this problem using a randomly chosen mapping called FrozenHopfield (FH) that is — due to its randomness — inherently unable to form meaningful abstractions. In fact, we show that under certain conditions, the representations are prone to collapse rendering the agent unable to distinguish between different inputs (see Fig. 1). We build upon HELM by (i) substituting FH with a feature-wise centering-and-scaling operation and (ii) incorporating a CLIP image encoder (Radford et al., 2021), which is pretrained in a multimodal fashion on web data consisting of images and text. We term the resulting new method HELMv2. Using HELMv2, we are able to distinguish even minute differences in the input when necessary, which drastically enhances downstream performance.

We demonstrate the effectiveness of HELMv2 on a diverse set of partially observable environments. Concretely, we train on 2D MiniGrid (Chevalier-Boisvert et al., 2018), and 3D MiniWorld environments (Chevalier-Boisvert, 2018). HELMv2 yields significant improvements over HELM in all environments. Further, we conduct ablation studies, which show that the improvements are not only due to the CLIP image encoder but also due to the replacement of the FH. Finally, we construct a mapping that successfully conveys the semantics extracted by CLIP to the LM. Surprisingly, however, using this mapping does not further improve the results in the selected environments.

## 2 Methods

The HELM framework consists of a history compression component, a learned CNN encoder of the current timestep, and an actor-critic head. The CNN and the actor-critic head are learned, while the history compression part is kept frozen. HELM has demonstrated that pretrained LMs are well suited for compressing past observations that are randomly mapped to language tokens. It performs two forms of compression: (i) spatial compression, and (ii) temporal compression. The former is realized with the FH mechanism, while the latter is performed with a pretrained TransformerXL (TrXL, Dai et al., 2019). The FH mechanism consists of a random matrix $\boldsymbol{P}$ and an attention mechanism over pretrained token embeddings (VocabAttn). More formally, let $\boldsymbol{E} = (\boldsymbol{e}_1, \ldots, \boldsymbol{e}_k)^\top \in \mathbb{R}^{k \times m}$ be the token embedding matrix of the pretrained LM consisting of $k$ embeddings $\boldsymbol{e}_i \in \mathbb{R}^m$. At every timestep $t$, we obtain inputs $\boldsymbol{x}_t \in \mathbb{R}^m$ for the LM from observations $\boldsymbol{o}_t \in \mathbb{R}^n$ via the FH mechanism by

$$\boldsymbol{x}_t^\top = \sigma(\beta \boldsymbol{o}_t^\top \boldsymbol{P}^\top \boldsymbol{E}^\top)\boldsymbol{E}, \tag{1}$$

where $\sigma$ is the softmax function and $\boldsymbol{P} \in \mathbb{R}^{m \times n}$ has entries sampled from $\mathcal{N}(0, n/m)$. The resulting $\boldsymbol{x}_t$ lies in the convex hull of the token embeddings of the LM. The parameter $\beta$ is a scaling factor that controls the dispersion of $\boldsymbol{x}_t$ within that convex hull.

**HELMcs** Our aim is to avoid representation collapse caused by the FH mechanism. We illustrate in Fig. 1 that FH is prone to representation collapse even for higher values of $\beta$, especially if observations tend to be visually similar. To sidestep this issue, we substitute the attention mechanism in FH with a

feature-wise centering-and-scaling operation. Let $\mathcal{B}$ be a batch of observations $\boldsymbol{o}_t$ and let $\boldsymbol{\mu}_{\mathcal{B}} \in \mathbb{R}^n$ be its mean feature vector and $\boldsymbol{\sigma}_{\mathcal{B}} \in \mathbb{R}^n$ the vector of standard deviations. Likewise let $\boldsymbol{\mu}_E \in \mathbb{R}^m$ and $\boldsymbol{\sigma}_E \in \mathbb{R}^m$ be means and standard deviations of the token embeddings $\boldsymbol{E}$. Then we compute $\boldsymbol{x}_t$ as

$$\boldsymbol{x}_t = \mathrm{diag}(\boldsymbol{\sigma}_E) \boldsymbol{P} \, \mathrm{diag}(\boldsymbol{\sigma}_{\mathcal{B}})^{-1}(\boldsymbol{o}_t - \boldsymbol{\mu}_{\mathcal{B}}) + \boldsymbol{\mu}_E, \tag{2}$$

where $\mathrm{diag}$ takes a vector and constructs a diagonal matrix from it. We refer to this setting as HELMcs.

**HELMv2**   We obtain another setting by encoding the observations with the ResNet-50 CLIP image encoder, which has the same output dimension as the embeddings $\boldsymbol{e}_i$ used by TrXL, thus eliminating the need for the random mapping $\boldsymbol{P}$. That is, we let $\boldsymbol{z}_t = \mathrm{CLIP}(\boldsymbol{o}_t) \in \mathbb{R}^m$ and construct $\mathcal{B}_\phi$ from $\boldsymbol{z}_t$. Consequently, $\boldsymbol{\mu}_{\mathcal{B}_\phi} \in \mathbb{R}^m$ and $\boldsymbol{\sigma}_{\mathcal{B}_\phi} \in \mathbb{R}^m$. We compute the LM inputs as

$$\boldsymbol{x}_t = \mathrm{diag}(\boldsymbol{\sigma}_E) \, \mathrm{diag}(\boldsymbol{\sigma}_{\mathcal{B}_\phi})^{-1}(\boldsymbol{z}_t - \boldsymbol{\mu}_{\mathcal{B}_\phi}) + \boldsymbol{\mu}_E. \tag{3}$$

The complexity imposed by the CLIP encoder is negligible since it is kept frozen and is only utilized during inference.

## 3   Experimental Results

We investigate the limitations of HELM on the MiniGrid-Memory environment (Memory, Fig. 1, left). The task for the agent is to memorize the object in the starting room and match it to the object at the end of the T-junction. If the agent chooses the wrong direction the episode ends without eliciting any reward. The objects only slightly differ in shape (green ball vs green key). We demonstrate that FH collapses to the same representation for both objects by measuring the distances $\delta$ between the softmax distribution $\sigma(\cdot)$ over token embeddings (Fig. 1, right). For very high values of $\beta > 1\mathrm{e}4$, the softmax converges to a one-hot encoding and $\delta$ directly corresponds to the probability of mapping to a single, but different token embedding. Therefore, with higher values of $\beta$, there is only a $10\%$ chance of avoiding collapse to the same token embedding. Moreover, we measure the cosine similarity between the embeddings of both observations for HELM, HELMcs, and HELMv2. Compared to HELM and HELMcs, HELMv2 is able to separate the two observations (see Table 1 in Appendix D). Moreover, we observe the random function $\boldsymbol{P}$ consistently conflates the inputs, while VocabAttn with high values of $\beta$ again separates them to some extent.

| Method | Cosine Similarity ($\downarrow$) |
|---|---|
| HELM, $\beta = 1$ | $0.99 \pm 1.68\mathrm{e}\text{-}7$ |
| HELM, $\beta = 10$ | $0.99 \pm 5.65\mathrm{e}\text{-}6$ |
| HELM, $\beta = 100$ | $0.99 \pm 8.2\mathrm{e}\text{-}4$ |
| HELM, $\beta = 1000$ | $0.97 \pm 0.08$ |
| HELMcs (ours) | $0.99 \pm 3.65\mathrm{e}\text{-}5$ |
| HELMv2 (ours) | $\mathbf{0.83 \pm 0.04}$ |

Table 1: Average cosine similarities between observation embeddings containing either key, or ball, for HELM, HELMcs, and HELMv2. Average is computed over batches of environment observations collected by a random policy.

We train all our methods with Proximal Policy Optimization (PPO, Schulman et al., 2017) on RGB observations. We conducted hyperparameter searches as mentioned in Appendix E and evaluate via the interquantile mean (IQM) and 95% bootstrapped confidence intervals (CIs) (Agarwal et al., 2021). To test for statistical significance, we perform a Wilcoxon test (Wilcoxon, 1945) at the end of training. We compare all our methods to an LSTM baseline on the Memory environment, the same set of partially observable MiniGrid environments as in (Paischer et al., 2022), and a set of eight more 3D MiniWorld environments (see Fig. 2). HELMv2 significantly outperforms HELM on MiniGrid ($p = 2.24\mathrm{e}\text{-}10$) and MiniWorld ($p = 0.012$). Also, HELMcs significantly outperforms HELM on the MiniGrid environemnts ($p = 8.7\mathrm{e}\text{-}4$). HELM, HELMcs and LSTM do not achieve performance better than randomly choosing a path at the end of the corridor for the Memory environment within 2M interaction steps. In contrast, HELMv2 is able to distinguish and memorize the objects and solves

the task. Additionally, we conduct an ablation study to isolate the different components of HELMv2 (see Appendix D). We find that the centering-and-scaling operation leads to much better results if performed in the abstract CLIP space, rather than in pixel space. Moreover, the choice of vision backbone affects final performance on the Memory environment.

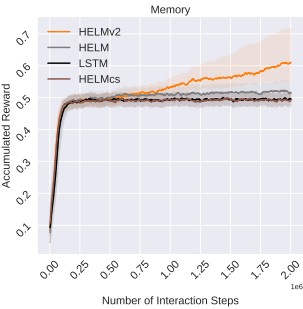
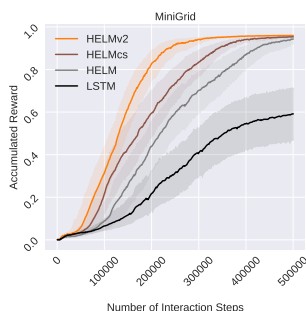
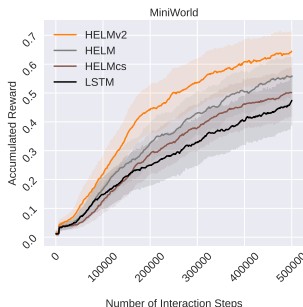

Figure 2: IQM and 95% bootstrapped CIs across 30 seeds on Memory environment (**left**), MiniGrid (**middle**), and MiniWorld environments (**right**).

## 4 Future Work and Discussion

In this work, we want to pave the way toward a semantic history compression for RL, that is to preserve the semantic concepts of an observation when mapping it to the LM space. We can compute such a mapping from the vocabularies of the CLIP language encoder $\mathcal{V}_{\text{CLIP}}$ and the LM encoder $\mathcal{V}_{\text{LM}}$ without access to additional data. First, we identify the overlap between the two vocabularies $\mathcal{V}_{OV} = \mathcal{V}_{\text{CLIP}} \cap \mathcal{V}_{\text{LM}}$ with size $l = |\mathcal{V}_{OV}|$. Next, we embed $\mathcal{V}_{OV}$ in the CLIP output space and the LM embedding space yielding embedding matrices $\boldsymbol{F} = (\boldsymbol{f}_1, \ldots, \boldsymbol{f}_l) \in \mathbb{R}^{n \times l}$ and $\boldsymbol{E}_l \in \mathbb{R}^{m \times l}$, respectively. We compute an orthogonal linear mapping matrix $\boldsymbol{W}$ according to the Procrustes method (Schönemann, 1966), as commonly used for aligning monolingual embedding spaces (Artetxe et al., 2018; Hoshen & Wolf, 2018; Smith et al., 2017; Lample et al., 2018; Zhang et al., 2016; Xing et al., 2015; Minixhofer et al., 2022). We refer to this setting as SHELM (for semantic HELM) and elaborate in more detail on it in Appendix B.

Due to the alignment of the image and text modalities in CLIP space, $\boldsymbol{W}$ can be used to project images to the LM space while preserving its semantics. We visualize sample images and the closest tokens in the LM space in Fig. 3. Further, in Appendix C we conduct a quantitative analysis using publicly available image captioning datasets and show that our mapping can identify tokens prevalent in captions provided by humans.

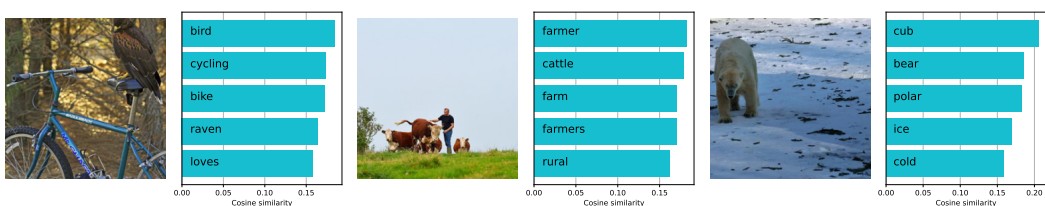

Figure 3: Top 5 closest tokens in the LM embedding space after applying our orthogonal mapping to an image embedded with CLIP.

We show preliminary results of SHELM in Fig. 7. Surprisingly, we observe performance on-par with HELMv2. We believe this is due to the fact that our selected environments are simulated 2D and 3D scenes for which the CLIP image encoder is unable to sufficiently extract semantically meaningful features. This is in line with findings of Fan et al. (2022) who minimally finetune CLIP for the Minedojo environment. We validate this finding by taking a closer look at the token rankings for the two observations of the Memory environment in Fig. 4 in the Appendix. However, we expect to see a difference in more realistic environments with observations closer to the CLIP pre-training domain (Tam et al., 2022). We aim to investigate this question in future work.

## Acknowledgements

The ELLIS Unit Linz, the LIT AI Lab, the Institute for Machine Learning, are supported by the Federal State Upper Austria. IARAI is supported by Here Technologies. We thank the projects AI-MOTION (LIT-2018-6-YOU-212), AI-SNN (LIT-2018-6-YOU-214), DeepFlood (LIT-2019-8-YOU-213), Medical Cognitive Computing Center (MC3), INCONTROL-RL (FFG-881064), PRIMAL (FFG-873979), S3AI (FFG-872172), DL for GranularFlow (FFG-871302), AIRI FG 9-N (FWF-36284, FWF-36235), ELISE (H2020-ICT-2019-3 ID: 951847). We thank Audi.JKU Deep Learning Center, TGW LOGISTICS GROUP GMBH, University SAL Labs initiative of Silicon Austria Labs (SAL), FILL Gesellschaft mbH, Anyline GmbH, Google, ZF Friedrichshafen AG, Robert Bosch GmbH, UCB Biopharma SRL, Merck Healthcare KGaA, Verbund AG, Software Competence Center Hagenberg GmbH, TÜV Austria, Frauscher Sensonic and the NVIDIA Corporation.

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

## A   Environments

We choose 8 diverse 3D environments of the MiniWorld benchmark suite:

- **CollectHealth:** The agent spawns in a room filled with acid and must collect medikits in order to survive as long as possible.
- **FourRooms:** The agent must reach a red box that is located in one of four interconnected rooms.
- **MazeS3Fast:** A procedurally generated maze in which the agent needs to find a goal object.

- **PickupObjs:** Several objects are placed in a large room and must be collected by the agent. Since the agent receives a reward of 1 for each collected object, the reward is unbounded.
- **PutNext:** Several boxes of various colors and sizes are placed in a big room. The agent must put a red box next to a yellow one.
- **Sign:** The agent spawns in a U-shaped maze containing various objects of different colors. One side of the maze contains a sign which displays a color in written form. The aim is to collect all objects in the corresponding color.
- **TMaze:** The agent must navigate towards an object that is randomly placed at either end of a T-junction.
- **YMaze:** Same as TMaze, but with a Y-junction.

We neglect the OneRoom and the Hallway environments, since those are easily solved by all our methods. Further, we neglect the Sidewalk environment since it is essentially the same task as Hallway with a different background. Since the reward of PickupObjs and CollectHealth are unbounded, we normalize them to be in the range of $(0, 1]$, which is the reward received in all other environments. For a more detailed description of the MiniGrid environments we refer the reader to Paischer et al. (2022).

# B   Linear Mappings between Language Embedding Spaces

Our aim is to create a linear mapping $\boldsymbol{W}$ from the aligned multimodal space of CLIP to the embedding space of a pretrained LM encoder. We argue that a linear mapping is sufficient since the semantic relation between the tokens is equivalent in both spaces and semantic relations between tokens can be expressed additively (Mikolov et al., 2013; Tewel et al., 2021). Therefore, given two CLIP tokens $\boldsymbol{f}_1, \boldsymbol{f}_2$, it does not matter whether we perform the addition before or after the mapping, i.e., $\boldsymbol{W}(\boldsymbol{f}_1 + \boldsymbol{f}_2) = \boldsymbol{W}\boldsymbol{f}_1 + \boldsymbol{W}\boldsymbol{f}_2$.

---

**Algorithm 1** Associating CLIP output space with LM input space

---

**Require:** CLIP language encoder $\text{CLIP}_{LM}$, Vocabulary of CLIP $\mathcal{V}_{\text{CLIP}}$, Language Model Embedding Layer $\text{LM}(v_i)$, Language Model vocabulary $\mathcal{V}_{\text{LM}}$
$\quad \mathcal{V}_{\text{OV}} \leftarrow \mathcal{V}_{\text{CLIP}} \cap \mathcal{V}_{\text{LM}}$        $\triangleright$ Search for overlapping vocabulary
$\quad \boldsymbol{f}_i = \text{CLIP}_{LM}(v_i) \quad \text{for} \quad v_i \in \mathcal{V}_{\text{OV}}$     $\triangleright$ Embed tokens in CLIP output space
$\quad \boldsymbol{e}_i = \text{LM}(v_i) \quad \text{for} \quad v_i \in \mathcal{V}_{\text{OV}}$      $\triangleright$ Embed tokens in LM input space
$\quad \boldsymbol{W} \leftarrow \texttt{create\_mapping}(\boldsymbol{F}, \boldsymbol{E})$     $\triangleright$ Compute mapping between embeddings

---

We consider all publicly available CLIP backbone variants and align their output spaces with the TrXL used in HELM. According to Algorithm 1 we determine the overlap in the CLIP and TrXL vocabularies. In principle any linear model can be applied to map from the CLIP space to the LM space.

We compare four different types of linear models:

- Ordinary Least Squares (Linear), minimizing

$$\arg\min_{\boldsymbol{W}} \|\boldsymbol{F}\boldsymbol{W} - \boldsymbol{E}\|_F^2 \qquad (4)$$

- Least Squares with Thikonov regularization (Ridge):

$$\arg\min_{\boldsymbol{W}} \|\boldsymbol{F}\boldsymbol{W} - \boldsymbol{E}\|_F^2 + \lambda\|\boldsymbol{W}\|_F^2 \qquad (5)$$

- Orthogonal Procrustes (Least Squares with orthogonality constraint, Schönemann (1966); Gower & Dijksterhuis (2005)):

$$\arg\min_{\boldsymbol{W}} \|\boldsymbol{F}\boldsymbol{W} - \boldsymbol{E}\|_F^2 \quad \text{subject to} \quad \boldsymbol{W}^\top\boldsymbol{W} = \boldsymbol{I} \qquad (6)$$

- Robust Procrustes (RobProc, (Groenen et al., 2005)) iteratively refines the Orthogonal Procrustes method based on the predicion error $\|\boldsymbol{F}\boldsymbol{W} - \boldsymbol{E}\|_F^2$[1].

---

[1]For the detailed algorithm, we refer the reader to Groenen et al. (2005)

In total, there are 5285 tokens that appear in both, $\mathcal{V}_{\text{CLIP}}$, and $\mathcal{V}_{\text{LM}}$. Prior work has found that as little as ten word correspondences are sufficient to train an orthogonal mapping between monolingual embedding spaces of closely related languages (Zhang et al., 2016). This assumes a certain degree of isomorphism between the embedding spaces. Since we train a mapping between embedding spaces of the same language, we expect this assumption to hold in our setting as well. The CLIP output space and the TrXL input space differ greatly in their statistics, therefore we perform centering and scaling as preprocessing. The Procrustes method is commonly used for aligning embedding spaces across different languages since it preserves *monolingual invariance* (Artetxe et al., 2016; Minixhofer et al., 2022). We consider various choices of CLIP visione encoders and align their output spaces with the TrXL used in HELM. We determine the intersection of the vocabulary of the CLIP language encoder and the TrXL language encoder.

We perform a 5-fold cross validation and measure the accuracy, considering each token as its own class. Table 2 shows the average train and test accuracy for the different linear mapping methods for various CLIP backbones and TrXL. We observe a strong overfitting effect for Linear, while the two Procrustes variants generalize best. We do not show variance estimates since those are negligibly small.

| | Linear | Ridge | Procrustes | RobProc |
|---|---|---|---|---|
| RN50 | 0.733/0.172 | 0.529/0.246 | 0.637/**0.289** | 0.658/0.285 |
| RN101 | 0.523/0.243 | 0.522/0.243 | 0.618/**0.304** | 0.65/0.303 |
| RN50x4 | 0.581/0.239 | 0.579/0.238 | 0.675/**0.319** | 0.701/0.309 |
| RN50x16 | 0.647/0.233 | 0.645/0.233 | 0.718/**0.332** | 0.742/0.33 |
| RN50x64 | 0.75/0.241 | 0.74/0.235 | 0.737/**0.342** | 0.752/0.34 |
| ViT-B/32 | 0.531/0.258 | 0.529/0.26 | 0.592/**0.308** | 0.616/0.3 |
| ViT-B/16 | 0.541/0.268 | 0.538/0.268 | 0.613/**0.329** | 0.638/0.327 |
| ViT-L/14 | 0.656/0.272 | 0.632/0.28 | 0.664/**0.351** | 0.683/0.346 |
| ViT-L/14* | 0.657/0.271 | 0.632/0.281 | 0.662/**0.353** | 0.68/0.348 |

Table 2: Train/Test accuracy for different linear models optimized for mapping CLIP tokens to the TrXL embedding space. Average over 5-fold cross validation is shown. ViT-L/14* received images resized to 336 pixels as input during pretraining.

## C   Mapping Visual Input to the Language Space

The prevalent alignment of modalities in the CLIP output space allows mapping visual inputs to the LM space. In turn we can substitute the FH mechanism in HELM with CLIP followed by our mapping $W$ to preserve the semantics of an observation in the LM space. This way, during inference in the RL experiments we can simply re-center and re-scale by the statistics of LM space to obtain a suitable input for the TrXL.

We conduct a quantitative analysis of the different mapping methods. In this regard, we use a pre-existing image/caption dataset to quantify how well the semantics of an image are preserved after mapping to the LM space. We draw a random subset of 1000 image-caption pairs of the popular MSCOCO dataset (Lin et al., 2014). The subset is filtered to contain only image-text pairs where the captions contain at least 5 tokens of our computed vocabulary overlap in Appendix B. For preprocessing of the captions we remove stop words and apply stemming to tokenized captions and tokens in the overlap. Next, we propagate the images through various CLIP backbones, to obtain an image embedding and map it to the LM space using our pre-computed mappings. Finally, we rank tokens in the LM space based on their cosine similarity to the mapped image. Based on the obtained ranking we compute the Mean Reciprocal Rank (MRR, Craswell, 2009) and the Normalized Discounted Cumulative Gain (NDCG, Järvelin & Kekäläinen, 2002).

Table 3 shows the NDCG for various CLIP backbones mapping to the embedding space of TrXL relative to ranking in the CLIP space (rNDCG). We observe that the Procrustes mapping consistently outperforms the Linear and Ridge mapping. Furthermore, there is no improvement in iteratively refining the Procrustes method as in RobProc. The results for MRR are shown in Table 4. Remarkably, on average the third ranked token is contained in a caption for the Procrustes mapping, while for CLIP

it is every second ranked token. The high variance stems from images that resulted in suboptimal rankings. From this analysis we conclude that most of the semantics are preserved with our linear mapping.

To validate that the CLIP image encoder fails to sufficiently extract semantics of environment observations, we take a closer look at token rankings for the observations in Fig. 1. Indeed, the tokens exhibiting the highest similarity in the CLIP space do not describe the semantics of the image, but rather similar concepts, i.e., the token *pong* is ranked higher when the ball is present in the observation.

| | Linear | Ridge | Procrustes | RobProc |
|---|---|---|---|---|
| RN50 | 0.636±0.17 | 0.701±0.18 | **0.774±0.171** | 0.772±0.172 |
| RN101 | 0.647±0.177 | 0.7±0.186 | **0.819±0.178** | 0.809±0.18 |
| RN50x4 | 0.638±0.164 | 0.68±0.172 | **0.824±0.178** | 0.814±0.165 |
| RN50x16 | 0.635±0.165 | 0.68±0.174 | **0.814±0.178** | 0.809±0.178 |
| RN50x64 | 0.632±0.167 | 0.647±0.165 | **0.797±0.179** | 0.791±0.178 |
| ViT-B/32 | 0.624±0.166 | 0.648±0.17 | **0.808±0.169** | 0.798±0.167 |
| ViT-B/16 | 0.61±0.158 | 0.663±0.176 | **0.822±0.186** | 0.81±0.187 |
| ViT-L/14 | 0.619±0.166 | 0.658±0.172 | **0.815±0.186** | 0.807±0.183 |
| ViT-L/14* | 0.61±0.164 | 0.646±0.172 | **0.81±0.188** | 0.802±0.186 |

Table 3: rNDCG for ranking of tokens in the LM space relative to ranking of tokens in the CLIP space. Image-caption pairs are drawn from the MSCOCO dataset. ViT-L/14* receives images resized to 336 pixels as input.

| | Linear | Ridge | Procrustes | RobProc | CLIP |
|---|---|---|---|---|---|
| RN50 | 0.036±0.108 | 0.107±0.225 | 0.229±0.323 | **0.232±0.336** | 0.514±0.4 |
| RN101 | 0.052±0.153 | 0.095±0.2 | **0.3±0.365** | 0.282±0.359 | 0.532±0.393 |
| RN50x4 | 0.03±0.082 | 0.074±0.179 | **0.308±0.37** | 0.284±0.353 | 0.524±0.391 |
| RN50x16 | 0.025±0.084 | 0.074±0.174 | **0.289±0.367** | 0.281±0.361 | 0.52±0.4 |
| RN50x64 | 0.052±0.158 | 0.063±0.174 | **0.285±0.362** | 0.276±0.355 | 0.532±0.396 |
| ViT-B/32 | 0.035±0.109 | 0.067±0.175 | **0.306±0.363** | 0.289±0.354 | 0.558±0.405 |
| ViT-B/16 | 0.038±0.136 | 0.092±0.208 | **0.349±0.378** | 0.332±0.379 | 0.554±0.4 |
| ViT-L/14 | 0.037±0.111 | 0.063±0.155 | **0.309±0.372** | 0.301±0.368 | 0.552±0.4 |
| ViT-L/14* | 0.039±0.11 | 0.065±0.155 | **0.327±0.376** | 0.318±0.371 | 0.575±0.393 |

Table 4: Mean Reciprocal Rank (MRR) for ranked tokens in the LM embedding space given an image as input and applying our mapping. Image-caption pairs are drawn from the MSCOCO dataset. MRR in the CLIP output space serves as upper bound. ViT-L/14* receives images resized to 336 pixels as input.

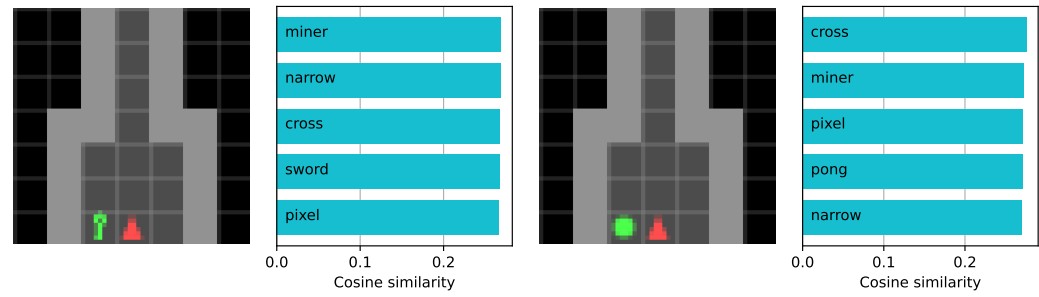

Figure 4: Top 5 closest tokens for observations of Memory environment in the CLIP space.

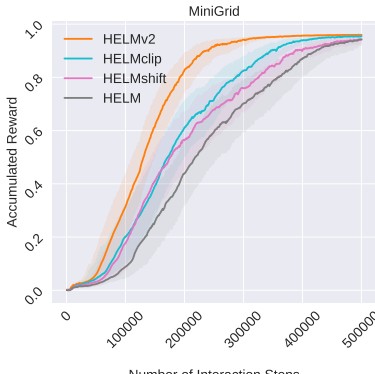
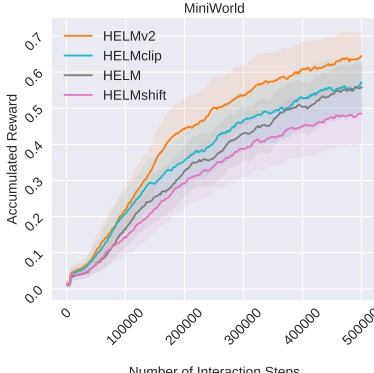

Figure 5: Mean IQM and 95% bootstrapped CIs across 30 seeds for RedBlueDoors (left), and TMaze (right) environments. HELMv2 consistently outperforms HELMclip on both environments.

# D   Additional Experimental Results

We conduct ablation studies to isolate the effect of the different components. In this regard, we add two additional settings: (i) HELMshift, and (ii) HELMclip. HELMshift adds the centering-and-scaling operation from HELMcs to the original HELM setting. HELMclip uses the same ResNet-50 CLIP image encoder with the centering-and-scaling operation followed by VocabAttn. We compare the performance of HELMclip and HELMshift to HELMv2 and HELM on all MiniGrid, and MiniWorld environments. Fig. 5 shows that adding the centering-and-scaling operation to HELM (HELMshift) does not lead to improved performance. In fact, on MiniWorld environments it even leads to worse performance than HELM. We suspect this is due to the frequent pixel changes in the observation space that do not correspond to significant changes in the environment state. While we also use image observations in MiniGrid the observation space is much more abstract since entire tiles change at once. Substituting $P$ in HELMshift with the CLIP encoder (HELMclip) however, results in an immediate improvement on MiniGrid environments. Since HELMclip still uses the VocabAttn, it is prone to representation collapse if changes in the CLIP space are subtle (see Fig. 1). Finally, discarding VocabAttn (HELMv2) drastically improves performance on both, MiniGrid and MiniWorld environment suites. While VocabAttn can help in some environments, its proneness to collapse overall hurts performance.

To show the effect of different image encoders, we perform an ablation where we substitute the CLIP image encoder with a ResNet (He et al., 2016, HELMv2-RN34-IN), and a Vision Transformer (Dosovitskiy et al., 2021, HELMv2-ViT-L16-IN), pretrained in a supervised manner on the popular ImageNet dataset (Deng et al., 2009). Additionally, we compare to a Vision Transformer version of CLIP (HELMv2-ViT-B/16). First, we measure cosine similarities to quantify the ability of the different vision encoders to separate the two observations (see Fig. 6, right). The HELMv2-ViT-B/16 variant exhibits the lowest cosine similarity between both observations. However, better separability does not correlate with improved downstream performance (see Fig. 6, left). In fact, the best performing agent uses the ViT-L16 pretrained on ImageNet. An explanation for this might be the difference in scale (Table 6) or the different pretraining paradigms (i.e. supervised vs. contrastive). We aim at answering this question in future work.

Next, we show additional results for the SHELM method which we introduced in Section 4. We select the best backbone-mapping combination by ranking all combinations according to the average absolute NDCG (aNDCG), and select the top 5 settings. Furthermore, we perform a Wilcoxon test for statistical significance between the combinations after bonferroni correction (Bonferroni, 1936). The two combinations, ViT-B/16+Procrustes, and ViT-L/14*+Procrustes, significantly outperform all competitors in terms of absolute NDCG. Due to the imposed complexity of ViT-L/14* (see Table 6), we choose ViT-B/16+Procrustes to instantiate SHELM. We show preliminary results of SHELM on the Memory environment, and our MiniGrid and MiniWorld environments in Fig. 7. Additionally, we compare with HELMv2+RandOrtho, which samples random orthogonal matrices from the Haar distribution (Stewart, 1980). Surprisingly, we observe no statistically significant differences in performance between the different methods.

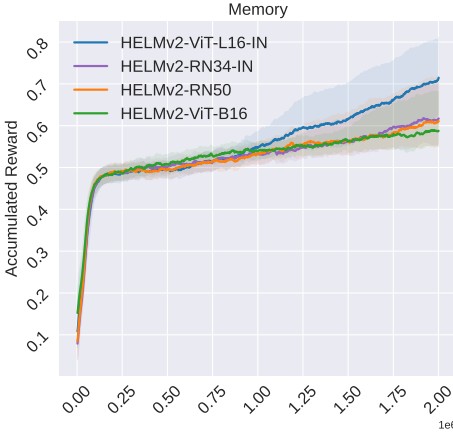

Figure 6: **Left:** IQM and 95% bootstrapped CIs across 30 seeds for Memory environment. **Right:** Average cosine similarity between observations containing a key and a ball for different vision encoders in HELMv2. Average is computed over batches of environment observations collected by a random policy.

| Method | Cosine Similarity ($\downarrow$) |
|---|---|
| HELMv2-RN50 | $0.83 \pm 0.04$ |
| HELMv2-ViT-B/16 | $\mathbf{0.73 \pm 0.06}$ |
| HELMv2-ViT-L16-IN | $0.83 \pm 0.04$ |
| HELMv2-RN34-IN | $0.76 \pm 0.04$ |

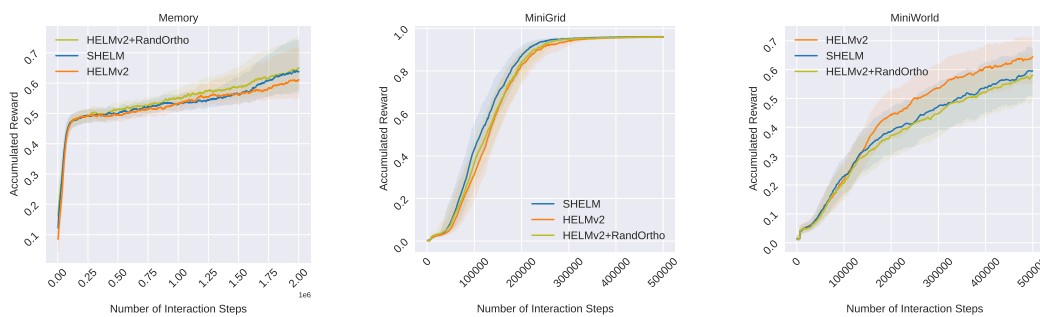

Figure 7: IQM and 95% bootstrapped CIs across 30 seeds on Memory (**left**), MiniGrid (**middle**), and MiniWorld (**right**) environments.

Finally, we conduct an additional analysis on how much the different methods preserve the semantics of natural images. In this regard, we perform the same quantitative analysis as in Appendix C for the different methods presented in this work. Again, we compute the MRR and the absolute NDCG for our MSCOCO subset of image-caption pairs. Results are shown in Table 5. We observe that methods such as HELMv2, or HELMv2+RandOrtho yield on-par or worse results compared to a random ranking. This is due to the fact, that for lower values of $\beta$ the VocabAttn tends to output the mean over token embeddings. In this case, the resulting ranking is equal across image, and thus, results in lower MRR and NDCG. Also substituting the RobProc mapping with a random mapping drawn from the Haar distribution leads to random rankings. Contrary, SHELM yields the best token rankings (see Table 3. Thus, for SHELM, we observe that most of the semantics are preserved. Adding VocabAttn after the RobProc mapping again results in loss of information which is mirrored in both MRR and NDCG.

## E   Hyperparameter Search

We search for hyperparameters for all methods that we trained and adapt the gridsearch conducten in Paischer et al. (2022). Particularly, we search for learning rate in $\{5e\text{-}4, 3e\text{-}4, 1e\text{-}5, 5e\text{-}5\}$, entropy coefficient in $\{0.05, 0.01, 0.005, 0.001\}$, rollout length in $\{32, 64, 128\}$ for HELMv2, and HELMcs. Since our analysis in Section 3 showed that for $\beta = 1, 10$ the spatial compression collapses to the mean over pretrained token embeddings, we alter the grid for $\beta$ of HELM to $\{100, 500, 1000, 5000\}$. To decrease wall-clock time of HELM variants, we vary the size of the memory register of TrXL such

|  | MRR (↑) | aNDCG (↑) |
|---|---|---|
| Random | 0.016±0.058 | 0.243±0.029 |
| HELMv2 | 0.018±0.057 | 0.244±0.031 |
| HELMv2+RandOrtho | 0.019±0.068 | 0.245±0.033 |
| SHELM+VocabAttn, $\beta = 1$ | 0.002±0.002 | 0.23±0.023 |
| SHELM+VocabAttn, $\beta = 10$ | 0.003±0.005 | 0.234±0.024 |
| SHELM+VocabAttn, $\beta = 100$ | 0.308±0.39 | 0.323±0.091 |
| SHELM+VocabAttn, $\beta = 1e3$ | 0.27±0.401 | 0.309±0.085 |
| SHELM+VocabAttn, $\beta = 1e4$ | 0.267±0.402 | 0.307±0.085 |
| SHELM | **0.349±0.378** | **0.35±0.088** |

Table 5: MRR and aNDCG for tokens ranked in the LM embedding space for different ablation setups.

| Vision Backbone | Approximate Parameter Count |
|---|---|
| CLIP-RN50 | 102M |
| CLIP-RN101 | 120M |
| CLIP-RN50x4 | 180M |
| CLIP-RN50x16 | 290M |
| CLIP-RN50x64 | 623M |
| CLIP-ViT-B/16 | 149M |
| CLIP-ViT-B/32 | 151M |
| CLIP-ViT-L/14 | 427M |
| CLIP-ViT-L/14* | 427M |
| RN34-IN | 21M |
| ViT-L/16-IN | 325M |

Table 6: Parameter count of different publicly available vision backbones used for HELMv2. ViT-L/14* receives images resized to 336 pixels as input.

that it can fit the maximum episode length (see Table 7). We lower the number of interaction steps for the gridsearch if we observe convergence before the 500k interaction steps. If no convergence is observed within the 500K interaction steps, we tune for the entire duration. We apply the same scheme for tuning the LSTM baseline and tune the same hyperparameters as in Paischer et al. (2022).

| Environment | Memory register of TrXL |
|---|---|
| DoorKey5x5 | 256 |
| DoorKey6x6 | 256 |
| DynamicObstacles | 64 |
| KeyCorridor | 256 |
| RedBlueDoors | 512 |
| Unlock | 256 |
| CollectHealth | 64 |
| FourRooms | 256 |
| MazeS3Fast | 256 |
| PickupObjs | 512 |
| PutNext | 256 |
| Sign | 32 |
| TMaze | 256 |
| YMaze | 256 |

Table 7: Length of memory register of TransformerXL used in HELM, HELMcs, and HELMv2 for selected MiniGrid (top) and MiniWorld (bottom) environments.

