# OpenReview forum: "Toward Semantic History Compression for Reinforcement Learning"
_NeurIPS.cc/2022/Workshop/LaReL — LaReL 2022_

### Official Review · Reviewer_9vgn · 2022-10-17
**Review for Towards Semantic History Compression for Reinforcement Learning**

**Rating:** 6
**Confidence:** 3

**Review:**

In this paper, the authors build upon a previous approach (History Compression via Language Models -- HELM) for creating long-term memories for agents faced with partially-observable, long horizon tasks. The authors demonstrate that the observation encoding used in HELM, based on a random encoding matrix followed by an attention mechanism, fails to discriminate between visually-similar but semantically different observations, which causes it to fail at certain tasks. The authors propose two alternative encoding schemes, the first one based on a center and scale transformation instead of the attention mechanism, and the second one based on the CLIP visual encoder. The authors demonstrate that both these encoding schemes improve on the baseline, and demonstrate that the one based on CLIP manages to solve the proposed tasks.

This paper supports its claims with empirical evidence and makes a good case that the proposed contributions are valuable for solving the tasks. The tasks are interesting and challenging and do indeed probe for the relevant capacity which is formation of long-term memory.

The motivation for the HELMcs encoding scheme doesn't seem very obvious to me, maybe an intermediate between HELM and HELM-CLIP would have been using pretrained image models instead of a random matrix ? In general it would be useful for the authors to elaborate a bit on the intuitions behind their proposed encoding methods.

In addition to that I would like to see reported the original results of agents with regular RNN-based hidden states. How do these agents perform on these tasks?

Overall the paper is sound and provides valuable additions to a memory compression scheme. The theme fits the workshop well, and I would like this work discussed at the workshop.

---

### Official Review · Reviewer_JUCv · 2022-10-18
**Interesting paper**

**Rating:** 7
**Confidence:** 4

**Review:**

# Summary

This paper studies the problem of compressing history for RL agents in memory-intensive tasks. It build on a recent method (HeLM) which shows that fixed pretrained LMs can be used to efficiently represent past state in RL agents, even in environments with non-linguistic state spaces, by simply randomly projecting states to combinations of fixed token embeddings via a frozen Hopfield network, without any fine-tuning or adaptation needed.

This paper identifies two improvements over the original HeLM work. The first aims to correct a deficiency that HeLM frozen hopfield network is prone to representational collapse because visually similar observations are often mapped to the same token embeddings; authors propose an alternative and perhaps simpler mapping that simply normalizes the state observations per batch and projects this to the token embedding space.

The second improvement is to start bringing pretrained semantics from image/language models into the history module. They do this by first by replacing the standard CNN encoder with a CLIP encoder, which (despite evaluation on artificial minigrid tasks) surely includes better visual representations (and indeed seems to work better). Second, they remove the random projection part, instead projecting CLIP observations in a way that actually preserves the original English semantics (semantic helm, or SHELM). This is a very obvious thing to do (since pretrained language models are also equipped to work with actual English semantics) but they observe that this does not result in any improvements (the authors call this surprising, but I honestly don't find it surprising at all given the environments).

# Strengths

- The suggested improvements indeed improve performance. A few additional environments are explored in the appendix which is nice, and verifies that we see improvements not just in the memory task (Which is perhaps adversarially designed for things like HeLM to fail, since there are visually similar observations with extremely different semantics)
- I find the semantic CLIP experiments at the end quite exciting, even if no positive results are observed. Yet I also find this particularly unsurpising given the simplicity of the MiniGrid environments and how far they are from really needing any rich semantics of the kind that CLIP would provide. Authors are spot on in L133 that semantics might really matter in more photorealistic environments (for some recent related work e.g. see https://arxiv.org/abs/2204.05080)

# Weaknesses

- The proposed way of incorporating a CLIP encoder leads to nice improvements but it may not be the most intuitive/practical decision in practice, since the CLIP model adds considerable complexity. Moreover, it begs comparison to an impotant baseline: if we happen to be using a CLIP encoder at each timestep to encode history, shouldn't we also use it in the agent itself, i.e. ? Is the more semantic representation offered by CLIP responsible for most of the gains we see with HeLMv2?
- There seems to be a missing baseline here, namely using CLIP as the observation encoder but using the orignial FH projection as in original HeLM? Again my concern is that CLIP semantics are primarily responsible for all of the gains here (e.g. that HeLM+CLIP performs identically to HeLMv2)

# Conclusion

Overall this is an interesting paper, I think it is missing some comparisons especially regarding the use of the CLIP models here, but I think this is a solid paper and would be happy to see it at the workshop.

# Questions

The proposed center-and-scaling operation is not intuitive to me. Could more be written about why this was chosen and how intuitively it should differ from the hopefield network projections? In particular the projection seems to depend on other elements in a batch of trajectories which is unintuitive (kind of like batchnorm). Why is this not a problem in practice?

Can you clarify whether the minigrid state representation is RGB or feature based? I assume RGB but it's not explicitly stated.

---

### Decision · Program_Chairs · 2022-10-21

Accept